# Nonalcoholic Fatty Liver Disease-Related Hepatocellular Carcinoma: The Next Threat after Viral Hepatitis

**DOI:** 10.3390/diagnostics13162631

**Published:** 2023-08-09

**Authors:** Mohamed Salaheldin, Heba Aly, Louis Lau, Shimaa Afify, Mohamed El-Kassas

**Affiliations:** 1Tropical Medicine Department, Faculty of Medicine, Ain Shams University, Cairo 11566, Egypt; drmstm81@yahoo.com (M.S.); drhebaismail105@gmail.com (H.A.); 2Department of Medicine and Therapeutics, Faculty of Medicine, The Chinese University of Hong Kong, Hong Kong 518172, China; louishslau@cuhk.edu.hk; 3National Hepatology and Tropical Medicine Research Institute, Cairo 11796, Egypt; 4Endemic Medicine Department, Faculty of Medicine, Helwan University, Cairo 11795, Egypt

**Keywords:** hepatocellular carcinoma, nonalcoholic fatty liver disease, nonalcoholic steatohepatitis, carcinogenesis

## Abstract

For many years, we have faced the complications of viral hepatitis and alcohol-related liver diseases such as cirrhosis, decompensation, portal hypertension, and hepatocellular carcinoma (HCC). Recently, we have seen a dynamic change in the field of hepatology. With the significant achievements in eradicating the hepatitis C virus by direct-acting antiviral agents and the rising epidemic of obesity, diabetes mellitus, and metabolic syndrome, there is a paradigm shift in the leading cause of liver cirrhosis and cancer to nonalcoholic fatty liver disease (NAFLD). Current data highlight the rapidly rising incidence of NAFLD-related HCC worldwide and expose the unseen part of the iceberg. In this review, we aim to update knowledge about the pathogenesis of NAFLD-induced HCC, surveillance difficulties, and promising disease markers. Molecular biomarkers, for example, may become a promising cornerstone for risk-stratified surveillance, early detection, and treatment selection for NAFLD-related HCC. Physicians can offer personalized and tailor-made clinical decisions for this unique patient subgroup.

## 1. Introduction

It is widely accepted that nonalcoholic fatty liver disease (NAFLD) is the major contributor to chronic liver diseases globally, especially after significant efforts against viral hepatitis, namely hepatitis B virus (HBV) vaccination and hepatitis C virus (HCV) treatment by direct-acting antiviral agents (DAA) [1]. NAFLD is defined as steatosis in 5% of hepatocytes with no other causes of fatty liver [2]. Defining the fatty liver patients at risk of disease progression is helpful. NAFLD prevalence increases at the same rate as the prevalence of obesity and type 2 diabetes mellitus (T2DM) worldwide [1]. Awareness of the NAFLD burden is important to show its effects on stopping disease progression [1,2]. NAFLD is a unique disease comprising a range of conditions extending from simple steatosis to steatohepatitis, fibrosis, and cirrhosis. The prevalence of NAFLD worldwide among adults is reaching about 25–30% [1]. The distribution differs between countries and regions, as the Middle East has the highest prevalence (32%), followed by South America (30%), with the least figures coming from Africa (13%); nevertheless, in developed countries, where the incidence of NAFLD is also increasing due to the increasing prevalence of the MetS and obesity [1,2]. The sedentary lifestyle and the growing prevalence of diabetes mellitus (DM), obesity, and metabolic syndrome have worsened the epidemic of NAFLD worldwide—an estimated prevalence of approximately 25%, and it can be as high as 60% to 70% in patients with DM. The epidemic of obesity can explain the rapid increase of NAFLD and NASH, type 2 DM, and other components of metabolic syndrome [2].

Hepatocellular carcinoma (HCC) is a common (the sixth leading cancer) and highly lethal malignancy (the third cause of cancer-related mortality) [3]. The most known causes of liver cancer are preventable and curable. The most common primary liver cancer globally is HCC, followed by cholangiocarcinoma. Liver tumors vary markedly according to risk factors and exposure. These risk factors include infections such as hepatitis B virus (HBV), hepatitis C virus (HCV), and liver flukes; behavioral factors (alcohol, tobacco); metabolic factors (metabolic syndrome); and aflatoxins [3]. And For decades, HBV, HCV, and alcohol were the main risk factors for cirrhosis and HCC. The annual incidence of HCC-related NAFLD (0.7% to 2.6% in patients with established cirrhosis and 0.1 to 1.3 per 1000 patient-years in patients with NAFLD but without cirrhosis) is lower than the incidence of HCC in patients with viral hepatitis (3–7% per year) [1,2]. However, considering that one-quarter of the adult population has NAFLD, we expect a sharp increase in NAFLD-derived HCC soon [4]. The pathological cascade contributing to the sequence from NASH to HCC is complex with various triggers, including excessive immune responses from cellular components such as endoplasmic reticulum and oxidative stress, organelle derangement, and DNA errors. Genetic and environmental factors can exaggerate those events [4]. NAFLD-HCC is usually detected at a later tumor stage and could also occur without preceding cirrhosis, but it also usually has a similar survival rate compared to HCV infection; this raises the suspicion of detecting cases that might require strict surveillance for early detection and effective treatment. As mentioned previously, it is alarming that up to 20–50% of NAFLD-derived HCC were found in patients with minimal or no fibrosis [5]. Among all causes of HCC, NAFLD-related HCC is the most rapidly growing indication for transplantation, reaching up to 16.2% of all transplantation cases in 2016 [6].

In a nationwide study from Italy, NAFLD was diagnosed in most patients with HCC (68.4%). The proportion of both total MAFLD and single etiology-NAFLD HCC significantly increased over time (from 50.4% and 3.6% in 2002–2003 to 77.3% and 28.9% in 2018–2019). Despite the late cancer stage at diagnosis, patients with NAFLD-induced HCC usually carry a lower risk of HCC-related death, denoting less aggressiveness of the tumor [7]. Moreover, it is estimated that NAFLD-related HCC represents about 34.8% of all HCC cases [8].

NAFLD-related HCC has some characteristic features that distinguish it from other etiologies such as HBV and HCV, which makes its diagnosis and management more complicated. As the patients are older, have more co-morbidities, and present at a more advanced stage. These characteristics adversely affect the prognosis and treatment. However, liver function is usually preserved in patients without cirrhosis. Interestingly, both diseases (NAFLD and HCC) are more common among males, which points to gender-specific factors such as sex hormones and lipid metabolism contributing to NAFLD-related HCC development [9].

From an epidemiological point of view, NAFLD-related HCC is a significant challenge, as 25% of the global population has NAFLD, and up to 25–50% of NAFLD-related HCC cases develop in non-cirrhotic patients [1,3]. The considerable disease burden is an emerging threat without an efficient and cost-effective screening method. As a result, NAFLD and NAFLD-related HCC have recently gained greater attention from hepatologists. In this review article, we aim to update knowledge about the pathogenesis of NAFLD-induced HCC, surveillance strategies, and potential biomarkers.

## 2. Pathogenesis: What Do We Know?

One of the obstacles to managing NAFLD-related HCC is the need for a better understanding of its complex pathogenesis [9]. Many driving and contributing factors interact in the development and progression of NAFLD-related HCC. This includes genetic and epigenetic drivers, systemic inflammation induced by obesity and insulin resistance, increased hepatic lipid storage and its consequences of lipotoxicity, endoplasmic reticulum stress, mitochondrial dysfunction, reactive oxygen species, and dysbiosis [10].

### 2.1. Genetic and Epigenetic Factors

Multiple genetic single nucleotide polymorphisms (SNPs) and epigenetic modifier dysregulation are identified in the pathogenesis of NAFLD-related HCC. However, the complex nature of NAFLD-related HCC pathogenesis and the role of nutritional and environmental factors may explain why none of these changes had a significant implication on clinical practice [9].

Recent data showed that, although many patients possess one of these SNPs, the majority do not develop HCC. It reflects the low positive predictive value of these SNPs. PNPLA3 rs738409, as an example, can confidently identify patients at lower risk for developing NAFLD-related HCC, not requiring close surveillance [11]. Moreover, combining multiple SNPs may give better predictive power for a future risk-stratified approach to identify patients carrying a higher risk of developing NAFLD-related HCC [12]. Table 1 shows the most studied genetic drivers identified in NAFLD-related HCC [13,14,15,16,17,18,19,20,21,22].

### 2.2. Epigenetics

DNA methylation dysregulation is a well-known factor in cancer development in different tumors [9]. As in the process of NAFLD development, the nuclear DNA and mitochondrial DNA (mtDNA) are affected due to fallacies in DNA methylation [10]. NAFLD-related HCC mouse models showed that hypomethylation associated with overexpression of the tubulin beta 2B class IIB (Tubb2b) gene is frequently noticed and is associated with HCC progression [23]. DNA methylation dysregulation may also represent a surrogate for assessing NAFLD severity by analyzing circulating cell-free DNA (cfDNA) methylation [24]. 

### 2.3. Non-Coding RNA (ncRNA)

The roles of long ncRNA and micro ncRNA (miRNA) in NAFLD-related HCC have been well studied recently, with multiple dysregulated ncRNAs involved. Non-coding RNA is an attractive field of research as it represents a potential diagnostic and therapeutic approach. They exert their effect by regulating gene expression and, hence, controlling signaling pathways [25]. Many long ncRNAs and micro ncRNAs (miRNAs) are associated with the development and progression of NAFLD-related HCC by affecting many signaling pathways, as shown in Table 2 [26,27,28,29,30,31,32,33,34].

### 2.4. Role of Gut Microbiome

Following the tight relationship between the gut and liver anatomically and functionally, the gut-liver axis may be involved in the progression of NAFLD into HCC. Dysbiosis potentially contributes to liver inflammation and NASH through gut permeability, lipopolysaccharide translocation, and immune activation [35]. However, patients with NAFLD-related HCC showed an increase in Bacteroides and Ruminococcaceae compared with those with NASH and NAFLD cirrhosis without HCC [35,36].

Dysbiosis-associated changes lead to an increase in secondary bile acids and a reduction in the expression of FXR (nuclear receptor) and disturb the lipid and carbohydrate metabolism associated with progressive liver disease. Pathogen recognition receptors (toll-like receptors) and macrophages induce liver inflammation, fibrosis, and proliferation, reducing antitumor immunity and causing chronic liver disease and carcinogenesis [37].

## 3. Surveillance and Diagnosis of NAFLD-Related HCC

### 3.1. Current Difficulties and Challengnes

The available evidence still suggests that NAFLD-related cirrhosis is a risk factor for HCC. Though this finding is suggested to be lower than that of HCV-related cirrhosis, the annual incidence of NASH cirrhosis is still higher than 1%. Nevertheless, HCC was also found in NAFLD patients without evidence of cirrhosis, with incidence rates lower than 1% per year. So, more high-quality prospective studies are needed to confirm these findings. This complex and incompletely understood pathogenesis represents significant differences in NAFLD-related HCC and impacts clinical practice. Guidelines recommend HCC surveillance in patients with advanced fibrosis/cirrhosis using abdominal ultrasound every six months [38]. Although this approach is widely accepted and simple, it needs more cost-effective evidence.

Recent data showed that the “one size fits all” approach is inappropriate, and tailored, individualized risk-based approaches should be implemented [39]. This should be highlighted more in patients with NAFLD due to its unique characteristics.

Twenty to fifty percent of HCC cases occur in non-cirrhotic patients with underlying NAFLD [4].

A significant burden on susceptible populations with NAFLD [1].Only 30% of patients with cirrhosis adhere to surveillance programs, and this ratio decreases among patients with NAFLD-related cirrhosis [40].Difficulties and limitations in ultrasound examination in patients with fatty liver disease, with up to 25% of ultrasound examinations being suboptimal [41]. Ultrasound alone has a very low sensitivity for detecting early HCC in cirrhotic patients. Although MRI has made remarkable progress in diagnosing HCC in NAFLD patients, the high cost and low availability still constitute a major limitation to its use in every patient [42].Variable natural history of NAFLD progression.

### 3.2. Current Techniques

The above difficulties emphasize the importance of using a risk-stratified approach in HCC surveillance, especially in patients with NAFLD. Molecular biomarkers for predicting progressive fibrosis and HCC development in patients with NAFLD are a hot topic with ongoing research. In the era of precision medicine, patients adopt personalized approaches based on their risks and characteristics. In NAFLD-derived HCC, this approach appears interesting, promising, and cost-effective [43]. In this study, it appeared that using abbreviated MRI in patients with a high risk of HCC development is more cost-effective than the universal approach using ultrasound. The advantages of MRI include its cost-effectiveness if compared to ultrasound regarding targeted diagnosis in high- and intermediate-risk patients due to marked recent reductions in its costs and also, compared to ultrasound, which is usually influenced by the operator’s skills and experience as well as the difficulties that can obstacle the diagnosis, especially in morbid obesity [8]. In morbid obesity, MRI yields more definite data that will be very important due to the increased and still increasing prevalence of obese patients with nonalcoholic fatty liver diseases. The advances of newly emerged simplified MRI protocols, such as AMRI, are expected to lower the bar for using MRI-based screening to save both money and time [44].

Moreover, MRI can measure liver fat percentage with the MRI proton density fat fraction (MRI-PDFF). Specifically, chemical shift-encoded (CSE) MRI is one of the best imaging indicators to detect early liver fat deposition [45]. Non-enhanced MRI is also a good option for HCC surveillance in high-risk patients. Another option is using a new diagnostic algorithm performed by the Japan Society of Hepatology for HCC in patients under surveillance for chronic liver disease, which contemplates the use of gadolinium-ethoxybenzyl-diethylenetriamine pentaacetic acid (EOB-MRI) [46,47]. Although expensive and time-consuming, MRI can measure liver fat percentage with magnetic resonance imaging proton density fat fraction (MRI-PDFF). Specifically, chemical shift-encoded (CSE) MRI is the best imaging indicator for early liver fat detection [48].

Nevertheless, patients who present with compensated advanced liver disease of NASH etiology, particularly obese patients with NASH, have a lower prevalence of portal hypertension compared with other etiologies [49].

Identifying high-risk patients among those with NAFLD includes identifying patients with a risk of NAFLD progression and advanced fibrosis/cirrhosis and identifying patients with NAFLD cirrhosis with a higher risk of HCC development. Currently, strategies for the identification of patients with a risk of NAFLD progression include:Using simple scores such as the Fibrosis-4 (Fib-4) score, the NAFLD fibrosis score (NFS), the Hepamet fibrosis score (HFS), and the enhanced liver fibrosis (ELF) panel. Among them, Fib-4 is easy and widely used. In a recent study comparing different non-invasive scores, HFS was the best performer for the identification of significant (F0–1 vs. F2–4, AUC = 0.758) and advanced (F0–2 vs. F3–4, AUC = 0.805) fibrosis, while NFS and FIB-4 showed the best performance for detecting histological cirrhosis (range AUCs 0.85–0.88) [50].Imaging techniques such as Fibroscan and magnetic resonance elastography.Combined imaging and laboratory results such as the FAST score [51].Liver biopsy is the gold standard for assessment, but due to its invasiveness, sampling errors, and inconvenience in follow-up, its role in clinical practice is limited.

### 3.3. What Is New?

N-terminal type III collagen propeptide (PRO-C3) is a marker for collagen formation associated with liver fibrogenesis. So, plasma PRO-C3 levels can correlate with the severity of steatohepatitis and the fibrosis stage. Moreover, the FIBC3 panel is an accurate method with a single threshold value for identifying F ≥ 3 fibrosis in NAFLD with maintained sensitivity and specificity and is superior to other commonly used non-invasive methods. ABC3D is a simplified version that is readily available for use in routine clinical practice. It has also been validated and shown similar accuracy. It can be assessed in serum using ELISA and has recently been used in two promising scoring systems: The ADAPT score [52] and FIBC3 [53]. Both showed promising results, better than Fib-4.

The GALAD score is a marker for HCC detection; however, recent data showed that the GALAD score was significantly higher 1.5 years earlier than HCC detection among patients with NAFLD who developed HCC than those who did not, highlighting its potential role as an HCC predictor. Moreover, the GALAD score is superior to individual serum markers for detecting HCC in NASH, regardless of the tumor stage or liver cirrhosis. This can suggest that GALAD should be investigated as a method for screening for NASH [54]. This approach was mainly based on identifying patients with advanced fibrosis and cirrhosis; however, the risk is not homogenous even in those patients. Non-invasive markers for fibrosis assessment in patients with NAFLD help discriminate between patients with advanced fibrosis and those without fibrosis, but their performance in predicting long-term HCC among patients with advanced fibrosis-cirrhosis is suboptimal [54]. Blood-based molecular biomarkers (PLSec) were recently identified for HCC prediction in patients with advanced fibrosis-cirrhosis [55]. PLSec showed good predictive ability for HCC in patients with advanced fibrosis and cirrhosis with different etiologies (adjusted hazard ratio [aHR], 2.35; 95% confidence interval [CI], 1.30–4.23). When combined with AFP, it showed higher predictive ability in two validation cohorts of patients with HCV cure after DAA (HR 3.8) and patients with HCC complete response and HCV cure (HR 3.08). Interestingly, PLSec-AFP performed significantly better in the subgroup of patients with NAFLD/cryptogenic cirrhosis (with an aHR of 11.9). PLSec-AFP is changeable with changes in risk factors such as HCV cure with DAA; thus, it has a potential role as a measurable method for estimating risk change with interventions for HCC prevention [55]. In this regard, validation of PLSec in NAFLD patients by a large-scale study will soon be a cornerstone of the change in practice (Figure 1).

### 3.4. Selection of Treatment Strategy

To date, different international guidelines do not consider HCC etiology in allocating patients to different therapeutic options, and BCLC is the most widely accepted staging system for HCC. According to the available evidence, NAFLD-related HCC might have a different treatment response than other etiologies. NAFLD-related HCC shows a similar response rate to liver transplantation, trans-arterial chemoembolization, ablation, selective internal radiation therapy, and tyrosine kinase inhibitors [56,57], but a better response to resection [58]. Regarding the surgery for resection, obesity should not be considered a risk factor. It was not associated with any increase in the operative time or increased bleeding. Moreover, laparoscopic liver surgery was feasible in obese patients, similar to lean patients. Increased postoperative morbidity rates were higher in obese patients, but they were mainly related to minor complications, such as abdominal wall infections; liver-specific complications and severe complications were not related to BMI. Good patient selection plays a key role in these circumstances, as in terms of liver function, liver volume, and associated co-morbidities, it is crucial to have good outcomes after liver surgery, independent of BMI [59,60].

Immune checkpoint inhibitors are approved as first-line therapy for advanced HCC (BCLC C). Surprisingly, patients with NAFLD-related HCC may have a limited response to immune checkpoint inhibitors compared to HCC due to other etiologies. Several agents of ICI have been approved as first- or second-line treatments for HCC. Analyses of survival outcomes in subgroups evaluating the efficacy of ICIs as first-line treatment revealed a discrepancy between viral HCC and non-viral HCC, including NASH-related HCC. Tremelimumab plus Durvalumab was the only combination of two ICIs tested against sorafenib in the first-line setting, and the only one that showed a relatively higher OS in the non-viral and viral HCC subgroups [61,62].

CD8 cells play a critical role in HCC development as they induce NASH-HCC rather than performing immune surveillance. Moreover, it was found that The HCC triggered by anti-PD1 (programmed death-ligand 1) was prevented by depletion of CD8 + T cells or TNF neutralization. Moreover, similar phenotypic and functional profiles in hepatic CD8 + PD1 + T cells were found in patients with NAFLD or NASH. A meta-analysis of three randomized clinical trials examined the inhibitors of PDL1 or PD1 in 1600 patients with advanced HCC, and their results showed that the immune therapy did not improve survival in those patients with non-viral HCC. Other studies showed that patients with NASH-driven HCC treated with anti-PD1 or anti-PDL1 showed poorer survival than patients with HCC due to other etiologies. Overall, these data show that NASH-HCC is less responsive to immunotherapy due to NASH-related abnormal T cell activation that induces tissue damage and leads to impaired immune surveillance [62].

Although these findings will require validation from larger-scale studies, they may highlight the importance of the etiology as a factor impacting the prognosis of HCC, and future selection of etiology-based therapeutic options may be anticipated.

### 3.5. Non-Pharmacological Prevention

HCC prevention mainly depends on managing modifiable risk factors such as viral hepatitis eradication and alcohol cessation. In NAFLD-derived HCC, modifiable risk factors such as DM, obesity, dysbiosis, disease activity, and fibrosis have been established [61]. The prevention strategy starts by adopting a change in multiple lifestyle factors, which can lead to a significant reduction in HCC incidence in general populations. This highlights the importance of promoting healthy living for the primary prevention of HCC [62]. Lifestyle interventions and chemoprevention are the main possible methods of HCC prevention in patients with NAFLD [63]. Weight loss (7% at least) benefits patients with NASH by improving inflammation, ballooning, steatosis, and activity scores [63]. Despite conflicting evidence regarding fibrosis regression, recent data suggested that 45% of studied NASH patients had fibrosis regression (in paired biopsies with 52-week intervals) after losing 10% of their body weight [64]. On the other hand, data directly linking weight loss and HCC prevention in patients with NAFLD is lacking. Physical exercise is associated with lower HCC risk independent of weight loss. This was proven by a recent multinational cohort study (HR 0.50; 0.33–0.76) in subjects performing vigorous physical exercise for 2 h twice per week [65]. The Mediterranean diet is widely recommended in patients with NAFLD due to its beneficial effect on lipid profile, glycemic control, cardiovascular risks, and liver fat contents [66]. Recently, a systematic review focusing on studies that link the Mediterranean diet and HCC risk showed that adherence to the Mediterranean diet is associated with a reduced risk of primary liver cancer [66]. The Mediterranean diet has been proven to benefit health, increase longevity, and decrease mortality. This can be performed by exerting anti-inflammatory effects, as it is deficient in saturated fat, refined sugar, and dairy and has a very high content of unsaturated fatty acids, fruits and vegetables, whole grains, and fish. This is similar to the foods consumed in Chinese culture, which are also associated with a lower HCC risk and may be due to the higher consumption of soy products, seafood, traditional soups, and herbal teas [67].

Multiple studies demonstrated a beneficial effect of regular coffee consumption in patients with chronic liver disease and NASH/NAFLD, decreasing HCC risk [68,69]. Although the exact amount is not well defined, a large meta-analysis showed a 35% reduction in the relative risk of HCC development in those who consume two cups of coffee daily [69]. However, the same meta-analysis failed to demonstrate a significant relative risk reduction in the case of decaffeinated coffee consumption [70].

### 3.6. Chemoprevention

Several drugs were associated with a reduced risk of HCC development in patients with NASH/NAFLD. Growing evidence supports the chemopreventive roles of aspirin, metformin, and statins. Regular administration of 650 mg of aspirin per week was beneficial in reducing 50% of HCC risk in a pooled analysis of two American cohort studies [71]. Long-term aspirin use was associated with a significant reduction in HCC risk, which is evident after five years of use and is also dose dependent. Those effects were not noted with non-aspirin NSAIDs [71]. This finding was confirmed in a Swedish study demonstrating a lower HCC risk (HR 0.69; 95% CI 0.62–0.76) in patients who used low-dose aspirin (less than 160 mg daily) for five years without a significant increase in GI bleeding risk [72]. Metformin is well known to have a chemopreventive role against the development of HCC in patients with DM, and this effect was confirmed in a meta-analysis and extensive population cohort studies. Metformin use in diabetic patients was also shown to lower HCC incidence by 41% to 78% compared to non-metformin users [73]. Many studies observed the protective role of statins in lowering HCC risk, which was confirmed by a meta-analysis showing a 46% decrease in HCC risk among statin users. It was found that lipophilic statins, such as lovastatin or simvastatin, are more beneficial than hydrophilic statins, such as pravastatin, for HCC. Moreover, it has shown better outcomes in the Western population than in the Asian population. Genetic variations in the structure or polymorphism between Asian and Western people may affect statins’ pharmacokinetics and pharmacodynamic properties [74]. Another meta-analysis confirmed this beneficial effect for lipophilic statins more than hydrophilic statins (51% versus 27%) [75].

The recent advances in molecular biomarkers as measurable predictors of HCC risk, such as PLSec AFP, may provide a method to guide future research by using them as an endpoint for preventive interventions. The blood-based PLSec-AFP sharply stratifies patients with advanced liver fibrosis who are at risk for long-term HCC and, therefore, gives a guide to risk-based, tailored HCC screening. This approach will identify higher-risk patients who may benefit more from interventional preventive measures. Moreover, this approach may offer a re-evaluation of individual risk after preventive interventions and the need to switch from one to another [55].

Radioembolization (RE) has been proven from retrospective studies to be effective for the treatment of locally advanced HCC with fewer side effects in comparison to systemic therapy. RE can be a safe option in the treatment strategy, especially in the case of potential contraindications for standard loco-regional and systemic treatments [56].

## 4. Conclusions

NAFLD has become the most common chronic liver disease and a serious issue for clinicians and healthcare systems around the globe as the incidence of overweight, obesity, insulin resistance, and T2DM has increased. Patients with NASH, severe fibrosis, or cirrhosis are at risk of developing complications and HCC. While liver biopsy is considered the gold standard for the diagnosis and assessment of NAFLD, it is not always a feasible option in clinical practice due to the large number of patients with NAFLD and the procedure’s invasiveness. Nonetheless, the level of liver fibrosis is the most important factor influencing liver-related morbidity and mortality, which can be assessed by different traditional and novel markers. Patients who have been diagnosed with cirrhosis should be closely monitored for HCC. Usually, standard screening tests will be utilized, including ultrasound and a serum AFP level every six months. The quality of the ultrasound will not be sufficient to confidently exclude hepatic focal lesions in a considerable percentage of individuals with NASH cirrhosis and/or obesity, rendering the potential role of MRI in this patient subgroup. Because of the unique characteristics of NAFLD-related HCC, the recent advances in molecular biomarkers may provide a target to guide future research by using them as a predictor for HCC and an endpoint for preventive interventions. Finally, growing evidence supports the chemo-preventive roles of aspirin, metformin, and statins in reducing the risk of HCC development among patients with NASH/NAFLD.

## Figures and Tables

**Figure 1 diagnostics-13-02631-f001:**
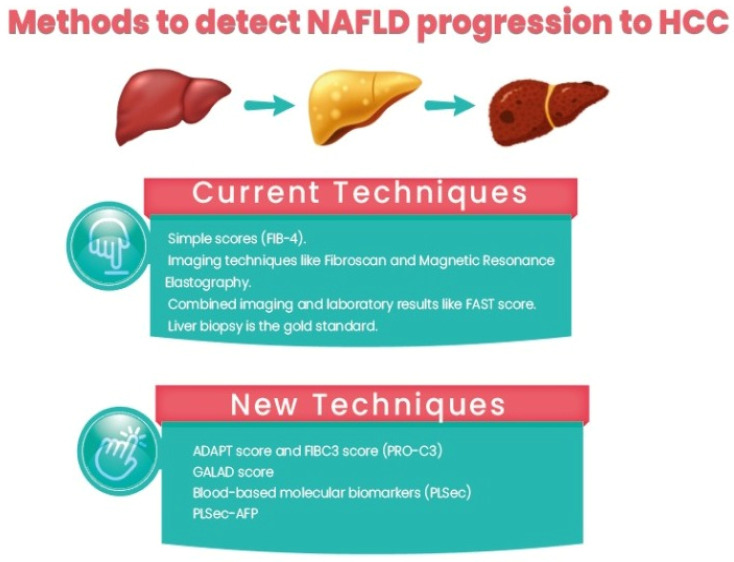
Management of NAFLD-related HCC.

**Table 1 diagnostics-13-02631-t001:** Single nucleotide polymorphism associated with NAFLD-related HCC.

Single Nucleotide Polymorphism	Association in NAFLD	Mechanism	Ref.
Patatin-like phospholipase domain containing 3 (PNPLA3) The rs738409 C > G single nucleotide polymorphism	increased risk of NASH and fibrosis and is an independent risk factor for the development of HCC	Accumulation of TG in hepatocytes HSCs activation Mitochondrial dysfunction	[12,13]
rs58542926 C > T missense variant in the Transmembrane 6 superfamily member 2 (TM6SF2) E167K	steatosis, inflammation, ballooning and fibrosis but it conferred protection against cardiovascular events Fibrosis and HCC (controversial)	impaired VLDL secretion and fat accumulation in hepatocytes	[14,15]
TM6SF2 silencing in HepG2 (TM6SF2−/−) by (CRISPR/Cas9)	NASH, fibrosis and, HCC	Mitochondrial dysfunction	[16,17,18]
rs641738 C > T variant in the TMC4/MBOAT7 locus	MBOAT7 rs641738 confers risk of fibrotic progression in NAFLD and independently associated with the development of HCC even in the absence of cirrhosis	impaired hepatic MBOAT7 function	[19,20]
The PDCD-1 gene encodes an inhibitory cell surface receptor involved in the regulation of T cell functions during immune responses/tolerance PDCD-1 rs7421861	NAFLD HCC	remodeling of the immune cell population	[21]

**Table 2 diagnostics-13-02631-t002:** Non-coding RNA associated with NAFLD-related HCC.

Dysregulated Non-Coding RNA	Association	Mechanism	References
Decreased levels of micro RNA122, 192 and 194 (in tissues)	Associated with NAFLD progression HCC Poor prognosis of HCC	Interfere with c-Myc pathway epithelial mesenchymal transition pathway	[25,26]
Micro RNA 21, 155, 375 and 16 in tissues and serum as cell free RNA	HCC development		[27]
Highly upregulated in liver cancer (HULC) long non-coding RNA	NAFLD progression HCC proliferation and metastases	Mitogen activated protein kinase (MAPK) signaling epithelial mesenchymal transition pathway	[28,29,30,31,32]
Metastases associated lung adenocarcinoma transcript 1 (MALAT1) long non-coding RNA	NAFLD fibrosis HCC development	Silent information regulator 1(SIRT1) and TGFbeta 1 Wnt signalling activation	[30,33]

## Data Availability

Not applicable.

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
