# Peer review of "Nonalcoholic Fatty Liver Disease-Related Hepatocellular Carcinoma: The Next Threat after Viral Hepatitis"

_diagnostics, 2023, doi:10.3390/diagnostics13162631_

Round 1
Reviewer 1 Report
The manuscript developed by your team and sent for publication highlights the work done to document and select the studies carried out up to this date and published in the specialized medical literature, studies that had as the main topic of study the NASH/NAFLD relationship - hepatocellular carcinoma.
The review presented by your team has interesting notes of originality but, unfortunately, it puts too much emphasis on some studies that are mentioned in the bibliography of the manuscript; studies that, in turn, conclude that additional studies are needed to demonstrate the pathogenic mechanisms involved in the evolution of NASH/NAFLD towards hepatocellular carcinoma.
Both in the specialized medical literature and in your manuscript, the methods by which the demonstration of this evolution and the discovery of the pathogenic link are attempted are, at the present moment, particularly sophisticated, expensive and applied more randomly on some hunches that things would be this way.
The necessity of these sophisticated and expensive methods shows at least two things: either the mechanisms unfold randomly and over a longer period of time, or the mechanism is much simpler and triggers very quickly, so that this moment cannot be captured by current exploration methods. These would probably be the explanations for the absence of biomarkers that are predictive for the evolution of NASH/NAFLD towards hepatocellular carcinoma.
Your manuscript tries and succeeds in shedding light on the methods by which the identification of evolutionary biomarkers is pursued, but for the current methods and for the current clinical practice, these biomarkers are far from being able to be established at the moment.
Author Response
We would like to thank the reviewers for their valuable comments and their sincere efforts and we have done the amendments as required point by point as follows:
Reviewer 1:
On behalf of all the contributing authors, I would like to express our sincere appreciations of your constructive comments concerning our article. These comments are all valuable and really highlights the topic very well. We appreciate the time and effort that you have dedicated for providing your valuable feedback on our manuscript. Lastly, we have been able to incorporate some changes to our manuscript to reflect most of the provided suggestions.
Reviewer 2 Report
TITLE
In my opinion the title is interesting and attractive.
KEYWORDS
Authors did not correctly report all keywords from MeSH Browser. In particular, for example, I checked “metabolic dysfunction-associated fatty liver disease,” on MeSH Browser and I did not find this KW. This is important, in my personal opinion, in order to increase the traceability of this paper (and consequently the possibility of the Journal to be cited by Readers and Stakeholders). I suggest the check of all KWs and to change the majority of those not reported on MeSH Browser (or introducing new KW from MeSH browser, without removing those now present).
ABSTRACT
The abstract is well structured and properly reflects the main text highlighting only the most important aspects of this paper, consequently no major adjustments are needed.
1. Introduction
The introduction is very interesting.
I suggest only to clarify the sentence: “NAFLD-related HCC has some characteristic features that distinguish it from other 84 etiologies like HBV and HCV, which makes its diagnosis and management more compli-cated.”. in particular, the authors can clarify “some characteristic features”.
2. Pathogenesis: What do we know?
It is very interesting section. However, I think that the readers of this journal are more interested in diagnosis. So, I suggest only a reduction of this section.
3. Surveillance and diagnosis of NAFLD-related HCC
Current difficulties
In my opinion, this section could be improved.
Could the authors discuss the following future scenario? In Europe, some countries in Asia and USA, the surveillance program and also the future strategy to improve the surveillance program [Non-enhanced magnetic resonance imaging as a surveillance tool for hepatocellular carcinoma: Comparison with ultrasound. J Hepatol. 2020;72(4):718-724. doi:10.1016/j.jhep.2019.12.001 ----- Proposal of a new diagnostic algorithm for hepatocellular carcinoma based on the Japanese guidelines but adapted to the Western world for patients under surveillance for chronic liver disease. J Gastroenterol Hepatol. 2016;31(1):69-80. doi:10.1111/jgh.13150], will allow to overcome the ultrasound limitations in the detections of HCC at an early and very early stages.
In fact, actually ultrasound identify 4 out 10 patients in very early or early stages [Surveillance Imaging and Alpha Fetoprotein for Early Detection of Hepatocellular Carcinoma in Patients With Cirrhosis: A Meta-analysis. Gastroenterology. 2018;154(6):1706-1718.e1. doi:10.1053/j.gastro.2018.01.064]. The first consequence will be the detection of even-increasing number of lesions in very early and early stage (small lesions). Could the Authors discuss these themes, cite the papers and report their ideas about these possible scenarios? For example: in cirrhotic patients with NAFALD, it is important to use MRI for studying the parenchyma, lesions and quantify the steatosis [Diagnostics (Basel). 2023;13(11):1852. Published 2023 May 25. doi:10.3390/diagnostics13111852].
Furthermore, it is possible that malignant lesions different from HCC could arise in the cirrhotic livers (and therefore in cirrhotic livers of patients with NAFLD); so, it is important to utilize the best imaging modality to explore di liver parenchyma such as MRI.
Please, could the authors discuss these themes, citing the above-mentioned papers?
Management of NAFLD-related HCC
Selection of treatment strategy
In my opinion, the authors must discuss also other treatment modalities such as, for example, radioembolization [Liver Cancer. 2019;8(6):491-504. doi:10.1159/000501484] or TACE alone or in association with immunotherapy [J Hepatocell Carcinoma. 2023;10:883-892. Published 2023 Jun 9. doi:10.2147/JHC.S404500], focusing on the fact that the toxicity profile of these patients is similar to those with different etiology (i.e. HBV).
In my opinion, it is important to write some lines concerning MRI PDFF (Quantitative MRI), citing [Diagnostics (Basel). 2023;13(11):1852. Published 2023 May 25. doi:10.3390/diagnostics13111852].
In my opinion, it is important to write some lines concerning the diagnosis of Portal Hypertension in these population [Liver Int. 2023;43(7):1446-1457. doi:10.1111/liv.15561 --- Am J Gastroenterol. 2022;117(11):1825-1833. doi:10.14309/ajg.0000000000001887].
REFERENCES
References do not reflect the style showed in the “Instruction for Authors”.
Author Response
Reviewer 2:
Thank you very much for your valuable comments and here are the amendments as required point by point;
TITLE
In my opinion the title is interesting and attractive.
KEYWORDS
Authors did not correctly report all keywords from MeSH Browser. In particular, for example, I checked “metabolic dysfunction-associated fatty liver disease,” on MeSH Browser and I did not find this KW. This is important, in my personal opinion, in order to increase the traceability of this paper (and consequently the possibility of the Journal to be cited by Readers and Stakeholders). I suggest the check of all KWs and to change the majority of those not reported on MeSH Browser (or introducing new KW from MeSH browser, without removing those now present).
Reply: Key words not present on MeSH are removed and replaced with others that are present on MeSH browser.
ABSTRACT
The abstract is well structured and properly reflects the main text highlighting only the most important aspects of this paper, consequently no major adjustments are needed.
Reply: Thanks for your comment really appreciated.
- Introduction
The introduction is very interesting.
I suggest only to clarify the sentence: “NAFLD-related HCC has some characteristic features that distinguish it from other 84 etiologies like HBV and HCV, which makes its diagnosis and management more complicated.”. in particular, the authors can clarify “some characteristic features”.
Reply: we added a clarification to the characteristics features with this sentence; As the patients are older, having more co-morbidities and present with a more advanced stage. These characteristics adversely affect the prognosis and treatment. However, the liver function is usually preserved in patients with no cirrhosis
- Pathogenesis: What do we know?
It is very interesting section. However, I think that the readers of this journal are more interested in diagnosis. So, I suggest only a reduction of this section.
Reply: This part was reduced.
- Surveillance and diagnosis of NAFLD-related HCC
Current difficulties
In my opinion, this section could be improved.
Could the authors discuss the following future scenario? In Europe, some countries in Asia and USA, the surveillance program and also the future strategy to improve the surveillance program [Non-enhanced magnetic resonance imaging as a surveillance tool for hepatocellular carcinoma: Comparison with ultrasound. J Hepatol. 2020;72(4):718-724. doi:10.1016/j.jhep.2019.12.001 ----- Proposal of a new diagnostic algorithm for hepatocellular carcinoma based on the Japanese guidelines but adapted to the Western world for patients under surveillance for chronic liver disease. J Gastroenterol Hepatol. 2016;31(1):69-80. doi:10.1111/jgh.13150], will allow to overcome the ultrasound limitations in the detections of HCC at an early and very early stages.
In fact, actually ultrasound identify 4 out 10 patients in very early or early stages [Surveillance Imaging and Alpha Fetoprotein for Early Detection of Hepatocellular Carcinoma in Patients With Cirrhosis: A Meta-analysis. Gastroenterology. 2018;154(6):1706-1718.e1. doi:10.1053/j.gastro.2018.01.064]. The first consequence will be the detection of even-increasing number of lesions in very early and early stage (small lesions). Could the Authors discuss these themes, cite the papers and report their ideas about these possible scenarios? For example: in cirrhotic patients with NAFALD, it is important to use MRI for studying the parenchyma, lesions and quantify the steatosis [Diagnostics (Basel). 2023;13(11):1852. Published 2023 May 25. doi:10.3390/diagnostics13111852].
Furthermore, it is possible that malignant lesions different from HCC could arise in the cirrhotic livers (and therefore in cirrhotic livers of patients with NAFLD); so, it is important to utilize the best imaging modality to explore di liver parenchyma such as MRI.
Please, could the authors discuss these themes, citing the above-mentioned papers?
Reply: Thank you for your valuable comment, the related informations were added with references.
Management of NAFLD-related HCC
Selection of treatment strategy
In my opinion, the authors must discuss also other treatment modalities such as, for example, radioembolization [Liver Cancer. 2019;8(6):491-504. doi:10.1159/000501484] or TACE alone or in association with immunotherapy [J Hepatocell Carcinoma. 2023;10:883-892. Published 2023 Jun 9. doi:10.2147/JHC.S404500], focusing on the fact that the toxicity profile of these patients is similar to those with different etiology (i.e. HBV).
- Reply: we added a paragraph about the RE option in the treatment of NAFLD associated HCC with the reference.
In my opinion, it is important to write some lines concerning MRI PDFF (Quantitative MRI), citing [Diagnostics (Basel). 2023;13(11):1852. Published 2023 May 25. doi:10.3390/diagnostics13111852].
- Reply: A paragraph about MRI PDFF was added with reference.
In my opinion, it is important to write some lines concerning the diagnosis of Portal Hypertension in these population [Liver Int. 2023;43(7):1446-1457. doi:10.1111/liv.15561 --- Am J Gastroenterol. 2022;117(11):1825-1833. doi:10.14309/ajg.0000000000001887].
- Reply: A paragraph about the diagnosis of Portal Hypertension was added with reference.
REFERENCES
References do not reflect the style showed in the “Instruction for Authors”.
Reply: References were reviewed to match the instruction for authors.
Round 2
Reviewer 1 Report
The review made by your team in accordance with the reviewers' indications is efficient and beneficial. The review, in its current form, is better structured and more valuable for readers.
However, I would like to convey to you the fact that I practice medicine every day at the patient's bedside and in the clinic where I work, many patients with NAFLD are hospitalized.
As a result of what is described in the review, I should expect that in 10 years, some of these patients will develop hepatocellular carcinoma!?. I don't think something like this can happen.
Anyway, through this review, I am warned that I could have such surprises, although I consider that a patient who ends up developing hepatocellular carcinoma as a result of NAFLD certainly goes through an intermediate state of modification of the liver architecture. It is impossible for a patient to pass directly from the NAFLD stage to the carcinoma stage without anatomic-pathological changes more advanced than those of steatosis.
Regarding performing magnetic resonance imaging on every patient with NAFLD, I think it is a utopia. Besides the fact that the diagnostic method is laborious and expensive, there is also the discomfort of some patients followed by the refusal to perform this investigation.
Author Response
Thanks a lot for your valuable comment:
We added a paragraph to emphasis on this point in the paragraph called “current difficulties and challenges”
- Difficulties and limitations in ultrasound examination in patients with fatty liver disease, with up to 25% of ultrasound examinations being suboptimal.41 ultra-sound alone has a very low sensitivity in detecting early HCC in cirrhotic patients. Although MRI made remarkable progress in diagnosing HCC in NAFLD patients, still the high cost and low availability constitutes a major limitation to MRI use in every patient.